# Novel Antibiotics for Gram-Negative Nosocomial Pneumonia

**DOI:** 10.3390/antibiotics13070629

**Published:** 2024-07-05

**Authors:** Maria Panagiota Almyroudi, Aina Chang, Ioannis Andrianopoulos, Georgios Papathanakos, Reena Mehta, Elizabeth Paramythiotou, Despoina Koulenti

**Affiliations:** 1Emergency Department, Attikon University Hospital, National and Kapodistrian University of Athens, 12462 Athens, Greece; mariotaalm@yahoo.gr; 2Department of Critical Care Medicine, King’s College Hospital NHS Foundation Trust, London SE5 9RS, UK; 3Department of Haematology, King’s College London, London SE5 9RS, UK; 4Department of Critical Care, University Hospital of Ioannina, University of Ioannina, 45110 Ioannina, Greece; 5Pharmacy Department, King’s College Hospital NHS Foundation Trust, London SE5 9RS, UK; 6School of Cancer & Pharmaceutical Sciences, Faculty of Life Sciences and Medicine, Kings College London, London SE5 9RS, UK; 7Department of Critical Care, Laikon Hospital, 11527 Athens, Greece; 8Antibiotic Optimisation Group, UQ Centre for Clinical Research, Faculty of Medicine, The University of Queensland, Brisbane 4029, Australia

**Keywords:** cefiderocol, ceftazidime–avibactam, ceftolazone–tazobactam, hospital-acquired pneumonia (HAP), imipenem–relebactam, meropenem–vaborbactam, multi-drug resistant (MDR), nosocomial pneumonia, sulbactam–durlobactam, ventilator-associated pneumonia (VAP)

## Abstract

Nosocomial pneumonia, including hospital-acquired pneumonia and ventilator-associated pneumonia, is the leading cause of death related to hospital-acquired infections among critically ill patients. A growing proportion of these cases are attributed to multi-drug-resistant (MDR-) Gram-negative bacteria (GNB). MDR-GNB pneumonia often leads to delayed appropriate treatment, prolonged hospital stays, and increased morbidity and mortality. This issue is compounded by the increased toxicity profiles of the conventional antibiotics required to treat MDR-GNB infections. In recent years, several novel antibiotics have been licensed for the treatment of GNB nosocomial pneumonia. These novel antibiotics are promising therapeutic options for treatment of nosocomial pneumonia by MDR pathogens with certain mechanisms of resistance. Still, antibiotic resistance remains an evolving global crisis, and resistance to novel antibiotics has started emerging, making their judicious use crucial to prolong their shelf-life. This article presents an up-to-date review of these novel antibiotics and their current role in the antimicrobial armamentarium. We critically present data for the pharmacokinetics/pharmacodynamics, the in vitro spectrum of antimicrobial activity and resistance, and in vivo data for their clinical and microbiological efficacy in trials. Where possible, available data are summarized specifically in patients with nosocomial pneumonia, as this cohort may exhibit ‘critical illness’ physiology that affects drug efficacy.

## 1. Introduction

Nosocomial pneumonia is one of the most common healthcare-associated infections [1]. It encompasses ventilator-associated pneumonia (VAP) and hospital-acquired pneumonia (HAP). Global guidelines are broadly consistent in their clinical criteria for diagnoses of HAP and VAP [2,3,4]. HAP is defined as pneumonia occurring ≥48 h after admission, which was not present at the time of admission to the hospital. VAP is pneumonia occurring 48–72 h after endotracheal intubation. Patients with HAP may deteriorate and, later on in their course, require intensive care unit (ICU) admission and mechanical ventilation (termed ventilated HAP).

Nosocomial pneumonia is associated with significant avoidable morbidity and mortality [2,3,4]. It is the leading cause of death from any nosocomial infection in critically ill patients, despite improvements in supportive therapy, preventative strategies, and advanced antimicrobial therapy. The microbial etiology of nosocomial pneumonia varies between regions and between hospitals, but currently, Gram-negative are the most frequently isolated pathogens [3,5,6].

Numerous multi-drug-resistant (MDR-), extensively drug-resistant (XDR-), and even pan-drug-resistant bacteria (PDR) have been identified to cause nosocomial pneumonia, with resistance patterns that are challenging to treat with current conventional antibiotics. These include extended-spectrum beta-lactamases (ESBLs) producing *Enterobacterales*, carbapenem-resistant *Enterobacterales* (CREs), and carbapenem-resistant *Acinetobacter* species that are designated as critical priority pathogens for the development of novel antibiotics and carbapenem-resistant strains of *P. aeruginosa* (CRPA) that are designated as high priority [2,7,8]. Infections caused by MDR pathogens are associated with delayed treatment, persistent or recurrent infection, prolonged hospital stay, and increased morbidity and mortality [9,10,11]. In addition, the conventional antibiotics required to treat MDR infections, e.g., polymyxins, often carry increased toxicity profiles [11]. There is, therefore, an urgent need to support the development of novel antimicrobial agents that target mechanisms of resistance, as well as clear guidance for antimicrobial stewardship to limit the emergence and spread of MDR pathogens [12].

In this narrative review, we critically and comparatively present the novel antibiotics that have been approved since 2018 for the treatment of nosocomial pneumonia caused by MDR Gram-negative pathogens including ceftolozane–tazobactam, ceftazidime–avibactam, meropenem–vaborbactam, imipenem–cilastatin–relebactam, cefiderocol, and sulbactam–durlobactam (Table 1). These antibiotics show promising in vitro and in vivo spectrums of activity, including activity against conventionally difficult-to-treat resistance (DTR) [13]. Gram-negative pathogens are not novel classes of antibiotics per se, but are largely beta-lactams (novel or existing), with the addition of beta-lactamase inhibitors (with the exception of cefiderocol) [14]. Tazobactam, avibactam, relebactam, and vaborbactam inhibit the activity of a number of beta-lactamases produced by Gram-negative bacteria and when combined with cephalosporins (ceftolazone, ceftazidime) or carbapenems (imipenem–cilastatin, meropenem), they expand their spectrum of activity by conferring them stability from hydrolysis by these enzymes. Durlobactam is a novel diazabicyclooctane beta-lactamase inhibitor, which inhibits class A, C, and D beta-lactamases and in combination with sulbactam is designated to treat *Acinetobacter* sp. infections. Cefiderocol binds iron and enters the bacterial cells through porin and iron transport channels. It is a poor substrate for the majority of beta-lactamases and is relatively unaffected by porin loss and efflux pump upregulation mechanisms of bacterial resistance [15]. We summarize and compare in vitro data for pharmacokinetics/pharmacodynamics (PK/PD) (Table 2), spectrum of antimicrobial activity, and known resistance patterns, as well as data from in vivo trials for their clinical and microbiological efficacy in their specified treatment setting of nosocomial pneumonia, especially in the intensive care unit (ICU; Table 3). Where possible, this article makes specific reference to ‘real world’ data on the use of these novel antibiotics in patients with nosocomial pneumonia. This is important, as critically ill patients often demonstrate altered ‘critical illness’ physiology, leading to differences in PK/PD compared to healthy study participants and implications for the antimicrobial efficacy and safety profile [16,17,18]. The ‘critical illness’ perturbances are diverse, and some of these physiological differences between normal alveoli and critical illness alveoli/pneumonia are demonstrated in Figure 1. For a drug to exert a therapeutic effect to treat lung infection, it must penetrate into the epithelial lining fluid (ELF) and achieve a concentration above the minimum inhibitory concentration (MIC) [16]. The degree of drug penetration is often expressed as a ratio of the area under the concentration–time curve (AUC) for ELF compared to plasma concentration (otherwise known as AUC in ELF-to-plasma ratio). Where provided by studies, we have reported these ratios as percentages (%) in our article.

Systemic inflammatory response syndrome (SIRS) in critical illness contributes to dysregulated fluid status. Fluid extravasation leads to increased interstitial fluid and ELF volumes, and the dilution of intrapulmonary concentrations of antibiotics [17,19].Increased pulmonary vascular endothelium and alveolar epithelium permeability in critical illness alters drug distribution and equilibration [17,20].Altered protein concentrations and hypoalbuminemia in critical illness lead to differences in protein binding of antibiotics. The alveolar membrane is relatively impermeable due to tight junctions, and therefore the passive diffusion of an antibiotic into the ELF is largely only possible if the drug is not protein-bound [16,17].Drug metabolism is altered in critical illness, by altered lymphatic drainage due to intrathoracic pressures in mechanically ventilated patients, and systematically altered excretion by the kidneys due to impaired renal function or supportive therapies including renal replacement therapy [17,19].

The figure was drawn by one of the authors on Adobe Fresco.

Drug penetration into ELF is determined by both drug-related and host-related factors. Drug-related factors include the degree of protein binding (as a largely unbound drug is able to diffuse between the tight joints of the alveolar membrane), hydrophilicity versus lipophilicity of the drug (which also determines its distribution in tissues), and other factors [16]. Host-related factors include permeability of the alveolar–capillary interface. For this reason, infection, inflammation, and immune dysregulation can significantly alter the permeability of the membrane and therefore drug penetration into the ELF (Figure 1) [16,19,20]. In Appendix A, dosing of novel antibiotics and MIC breakpoints are depicted.

Whilst the novel antibiotics summarized in our article represent promising efficacious options for treatment of certain strains of MDR nosocomial pneumonia, antimicrobial resistance remains an evolving global emergency. In this article, we discuss specific examples of resistance mechanisms to these agents that have been discovered since their introduction. Further study into these resistance mechanisms is paramount, to guide further antibiotic development and stewardship.

**Table 2 antibiotics-13-00629-t002:** Pharmacokinetic and pharmacodynamic parameters of ceftazidime–avibactam, meropenem–vaborbactam (RPX7009), ceftolozane–tazobactam, imipenem–cilastatin–relebactam (MK-7655), cefiderocol, and sulbactam–durlobactam (ETX2514) from respective clinical trials.

Author (Year), Trial Type, Dose and Regimen	Population Included Age of Participants (In Years)	CrCl (mL/ min)	Antibiotic	AUC (ELF/ Total Plasma; AUC_E/TP_) Concentration Ratio	AUC (ELF/ Free Plasma; AUC_E/FP_) Concentration Ratio	Cmax (Plasma)	Cmax (ELF)	Tmax (Plasma)	Tmax (ELF)	t½ (Plasma), Hours
**Nicolau (2015) [21]**. **Trial**: Phase 1, open-label study, healthy subjects. **Regimen**: 3 days (9 doses) of IV ceftazidime–avibactam q8 h at - 2000/500 mg; - 3000/1000 mg.	n = 43 healthy adult subjects (age 18–50). **Mean age ± SD:** - For 2000/500 mg: 30 ± 8 - For 3000/1000 mg: 33 ± 9	Not provided		**Mean AUC_E/TP_**	**Mean AUC_E/FP_**	**Mean (mg/L; variation %)**	**Mean (mg/L)**	**Median (range), hours**	**Median (hours)**	**Mean ± SD (hours)**
			**Ceftazidime**	Overall: 31–32% 2000 mg: 31.3% 3000 mg: 32.4%	Not provided	2000 mg: 90.1 (13.3) 3000 mg: 140 (9.6)	2000 mg: 23.2 3000 mg: 32.7	2000 mg: 2.00 (1.97–2.02) 3000 mg: 2.00 (1.97–2.03)	2 h	2000 mg: 2.86 ± 0.294 3000 mg: 2.94 ± 0.318
			**Avibactam**	Overall: 32–35% 500 mg: 34.9% 1000 mg: 32.0%	Not provided	500 mg: 14.5 (9.7) 1000 mg: 28.5 (10.2)	500 mg: 5.1 1000 mg: 7.9	500 mg: 2.00 (1.97–2.02) 1000 mg: 2.00 (1.97–2.03)	2 h	500 mg: 3.29 ± 0.82 1000 mg: 3.61± 0.47
**Wenzler (2015) [22]**. **Trial**: Phase I, randomized, open-label, multiple-dose study. **Regimen**: 3 doses of 2 g IV meropenem and 2 g IV vaborbactam (RPX7009) q8 h.	n = 25 healthy adult males and females (18–55 y). **Mean age ± SD**: 39.0 ± 10.6	Mean (SD): 93.9 ± 22.8		**Mean AUC_0–8, E/TP_**	**Mean AUC_0–8, E/FP_**	**Mean ± SD (μ g/mL)**	**Mean ± SD** **(μ g/mL)**	** **Mean (hours)** ^Φ^ **	** **Mean (hours)** ^Φ^ **	**Mean ± SD (hours)**
			**Meropenem**	63%	65%	58.2 ± 10.8	28.3 ± 6.69 *	3.25 ^Φ^	3 h ^Φ^	1.03 ± 0.15
			**Vaborbactam (RPX7009)**	53%.	79%	59.0 ± 8.4	26.1 ± 7.12 *	3.25^Φ^	3 h^Φ^	1.27 ± 0.21
**Rizk (2018) [23].** Trial: Phase I open-label, randomized, parallel-group study. **Regimen**: 5 doses of IV relebactam with imipenem–cilastatin, 250 mg/500 mg, q6 h.	Study group: n = 17 healthy adult subjects (n = 16 completed the study). **Age range**: 24–42 years	All patients had CrCl ≥ 80		**Mean AUC0_0-∞;E/TP_**	**Mean AUC0_0-∞;E/FP_**	**Mean (μM)**	**Mean (μM)**	**Mean (hours)**	**Mean (hours)**	**Mean (hours)**
			**Imipenem–cilastatin**	44.2%	55.2%	99.6 μM	32.6 μM	0.5	0.5	Plasma—0.95 ELF—1.03
			**Relebactam (MK-7655)**	43.0%	53.7%	47.9 μM	15.3 μM	0.5	0.5	Plasma—1.24 ELF—1.29
**Katsube (2019) [24].** **Trial**: Phase I, single-center, open-label study. **Regimen**: A single dose of 2 g IV cefiderocol.	n = 20 healthy adult male subjects aged 20 to 40 years. **Mean age ± SD**: 26 ± 4	Mean ± SD: 123 ± 16		**Mean AUC_0–6_, _E/TP_**	**Mean AUC_0–6, E/FP_**	**Mean (mg/L)**	**Mean (mg/L)**	**Median (hours)**	**Median (hours)**	**Mean (hours)**
			**Cefiderocol**	10.1%.	23.9%	142 mg/L	13.8 mg/L	1.0	1.0	Plasma—1.79 h ELF—1.76 h
**Caro (2020) [25].** **Trial**: Open-label, multicenter phase 1 trial in patients with VAP. **Regimen**: 4–6 doses of IV 3 g ceftolozane–tazobactam (2 g/1 g) q8 h. Doses adjusted for renal function.	n = 26 critically ill mechanically ventilated adult patients (>18 y) in intensive care, with proven/ suspected pneumonia. **Mean age ± SD:** 63.0 ± 16.3	Mean ± SD: 121.5 ± 76.6 mL/min		**AUC_0–8, E/TP_**	**AUC_0–8; E/FP_**	**Cmax (plasma)**	**Cmax (ELF)**	**Median (range), hours**	**Median (hours)**	**Mean (variation; %), hours**
			**Ceftolozane**	Not provided	50%	Not provided	Not provided	After first dose: 1.00 (0.88–2.00) After last dose: 1.00 (0.92–2.17)	6	After first dose: 4.15 (56.1) After last dose: 4.86 (61.5)
			**Tazobactam**	Not provided	62%	Not provided	Not provided	After first dose: 1.00 (0.88–1.12) After last dose: 1.00 (0.92–2.17)	2	After first dose: 2.15 (56.4) After last dose: 2.33 (49.7)
**Rodvold (2018) [19].** **Trial**: Phase 1, multiple-dose, open-label pharmacokinetic study in healthy adult male and female subjects. **Regimen**: 3 doses of IV 1 g durlobactam (ETX2514) and 1 g sulbactam, q6 h.	n = 30 healthy adult subjects. **Mean age ± SD:** 42 ± 11	Mean ± SD: 110 ± 22		**AUC_0–6_, _E/TP_ ratio (mean/median)**	**AUC_0–6; E/FP_ (mean/median)**	**Mean ± SD (mg/L) after 3rd dose**	**Mean ELF Cmax (mg/L)**	**Mean (range), hours**	**Mean (range), hours**	**Mean ± SD (hours)**
			**Sulbactam**	Mean: 50% Median: 50%	Mean: 81% Median: 80%	23.10 ± 7.61	Not provided	2.56 (2.0 to 3.05) hours	Not provided	Plasma: 1.12 ± 0.14
			**Durlobactam (ETX2514)**	Mean: 37% Median: 36%	Mean: 41% Median: 40%	33.41 ± 8.99	Not provided	2.62 (2.0 to 3.05) hours	Not provided	Plasma: 1.40 ± 0.18

IV: intravenous; q6 h/q8 h: every 6 h, every 8 h; population (n = number of participants); age (in years): mean ± standard deviation (SD); CrCl: creatinine clearance (mL/min) for study population; ELF: epithelial lining fluid; AUC: area under the concentration–time curve (%), and time frame and type (median/median) specified; AUC_E/TP_ ratio: AUC ELF-to-total plasma concentration ratio (total concentration of drug in plasma including protein-bound and unbound drug); AUC_E/FP_: AUC ELF-to-free plasma (unbound drug concentration accounting for protein binding) ratio, and this is a marker of intrapulmonary penetration of the drug. Cmax: peak plasma concentration (units are provided); Tmax: time taken for drug to reach Cmax; t½; elimination half-life (in hours). ^Φ^ Tmax was not explicitly stated, and was extracted from the graph and table of mean concentrations. * Cmax in ELF was not explicitly stated, so the greatest concentration (at Tmax) was extracted from the table of mean concentrations.

**Table 3 antibiotics-13-00629-t003:** Summary of trial demographic data and subgroup data on study participants admitted to the intensive care unit (ICU) at randomization.

Author (Year)	Trial	Antibiotic and Regimen *	Study Region and Study Date	Treatment Setting and Pathogens	Initial Randomized Sample Size	Subgroup for Which Data Extracted	Age of Participants in Subgroup	ICU Admissions (%) and APACHE II at Randomization in Subgroup	Efficacy Outcome
**Kollef (2019) [26]**.	**ASPECT-NP Trial:** Randomized, controlled, double-blind, phase 3, non-inferiority trial	IV 3 g CEFT-TAZ q8 h for 8–14 days versus IV 1 g MERO q8 h for 8–14 days.	263 hospitals in 34 countries in Europe, Australia, New Zealand, North America, South America, Asia/Pacific, Africa. **Study date**: Jan 2015–April 2018.	Hospitalized adults (≥18 years), who were intubated and mechanically ventilated, and had VAP or ventilated HAP. **Pathogen**: Gram-negative (monomicrobial and polymicrobial).	n = 726 randomized, of which CEFT-TAZ: n = 362 (49.9%) MERO, n = 364 (50.1%)	**ITT population ^1^, n = 726** CEFT-TAZ: n = 362 (49.9%) MERO, n = 364 (50.1%)	**Mean age (years) ± SD:** CEFT-TAZ: 60.5 ± 16.7 MERO: 59.5 ± 17.2	**Admitted to ICU:** n = 668/726 (92%) **APACHE II; Mean ± SD** CEFT-TAZ: 17.5 ± 5.2 MERO: 17.4 ± 5.7	CEFT-TAZ was non-inferior to MERO in terms of 28-day all-cause mortality and clinical cure at test-of-cure.
**Torres (2018) [27].**	**REPROVE Trial**: Multinational, phase 3, double-blind, non-inferiority trial	IV 2 g/500 mg CEF-AVI q8 h versus IV 1 g MERO q8 h for 7–14 days.	136 centers in 23 countries (Europe, Asia, South America, Asia, Africa). **Study date**: April 2013–Dec 2015.	Hospitalized adults (18–90 years), with nosocomial pneumonia including VAP and non-VAP (all patients with nosocomial pneumonia who did not have VAP). **Pathogen**: Gram-negative (monomicrobial and polymicrobial).	n = 879 randomized (of which 62 were excluded, leaving n = 817): CEF-AVI: n = 409 (50%) MERO: n = 408 (50%)	**Clinically modified-ITT population (cMITT) ^2^, n = 726:** CEF-AVI, n = 356 (49%) MERO, n = 370 (51%)	**Mean age ± SD**: CEF-AVI: 62.1 ± 16.6 MERO: 62.1 ± 16.6	**Mechanically ventilated ^Ƭ^:** n = 313/726 (43%) **APACHE II; Mean ± SD** CEF-AVI: 14.5 ± 4.01 MERO: 14.9 ± 4.05	CEF-AVI was non-inferior to meropenem in the treatment of nosocomial pneumonia.
**Wunderink (2018) [28].**	**TANGO II Trial: ** Phase 3, multinational, open-label, randomized controlled trial	IV 2 g/2 g MERO-VAB q8 h for 7–14 days versus BAT ^Φ^.	Multinational study across North America, Europe, Asia/Pacific, South America (and others). **Study date**: 2014 to 2017.	Hospitalized adults (>18) with confirmed or suspected CRE infections including complicated UTI, acute pyelonephritis, complicated intra-abdominal infection, HABP/VABP, and CRE bacteremia. **Pathogen**: Gram-negative CREs including *K. pneumoniae*, *E. coli*, *E. cloacae* sp., *P. mirabilis*, and *S. marcescens*.	n = 77 randomized, of which MERO-VAB, n = 52 (67.5%) BAT, n = 25 (32.5%)	**Microbiologically confirmed CRE infection (mCRE-MITT) ^3^: n = 47** MERO-VAB, n = 32 (68.1%) BAT, n = 15 (31.9%)	**Mean age ± SD:** Overall: 62.5 ± 13.7 MERO-VAB: 63.5 ± 14.1 BAT: 60.2 ± 13.0	**Admitted to ICU**: n = 8/47 (17%) **APACHE II**: no data provided	MERO-VAB monotherapy for CRE infection was associated with increased clinical cure, decreased mortality, and reduced nephrotoxicity compared with BAT.
**Motsch (2020) [29].**	**RESTORE-IMI 1 Trial:** Phase 3, randomized, double-blind study	IV 500/250 mg IMI/REL q6 h versus colistin (IV loading dose, followed by maintenance doses q12 h), plus IMI (500 mg q6 h), 5–21 days.	16 sites from 11 countries in South America, Europe, USA, and Turkey. **Study date**: October 2015 and September 2017.	Hospitalized adult (>18) patients with imipenem-non-susceptible (but colistin- and IMI/REL-susceptible) Gram-negative bacterial infections, including HAP, VAP, cUTIs, or cIAIs. **Pathogen**: Gram-negative pathogens including *P. aeruginosa*, *K.* spp., *C. freundii*, *and other Enterobacteriaceae*.	n = 47 randomized, of which IMI/REL, n = 31 (66%) Colistin–imipenem, n = 16 (34%)	**Microbiologic modified ITT (mMITT) ^4^, n = 31:** IMI/REL, n = 21 (67.7%) Colistin–imipenem, n = 10 (32.3%)	**Median age (range):** Overall: 59 (19–80) IMI/REL: 59 (19–80) COL + IMI: 61 (49–80)	**Admitted to ICU**: no data provided **APACHE II score, Median (range) ** HAP/VAP: 18 (0, 26) cIAI: 16 (14, 19) cUTI: 5.5 (0, 17)	Imipenem–relebactam is an efficacious and well-tolerated treatment option for carbapenem-non-susceptible infections.
**Titov (2021) [30].**	**RESTORE-IMI 2 Trial:** Phase 3 randomized, double-blind, non-inferiority trial	IV 500/500/250 mg IMI/REL at q6 h for 7–14 days versus IV 4 g/500 mg PIP/TAZ q6 h for 7–14 days.	113 hospitals in 27 countries (Americas, Europe, Asia, Australia). **Study date:** January 2016 and April 2019.	Hospitalized adult (>18) patients with nonventilated HABP, ventilated HABP, or VABP. **Pathogen**: Gram-negative pathogens (*K. pneumoniae*, *P. aeruginosa*, *Acinetobacter calcoaceticus–baumannii complex*, *E. coli).*	n = 537 randomized, of which IMI/REL, n = 268 (49.9%) PIP/TAZ, n = 269 (50.1%)	**Modified ITT population (MITT) ^5^, n = 531:** IMI-REL, n = 264 (49.7%) PIP-TAZ, n = 267 (50.3%)	**Mean age ± SD:** IMI/REL: 60.5 ± 16.9 PIP/TAZ: 58.8 ± 18.4	**Admitted to ICU:** **n = 351/531** (66.1%) **APACHE II; Mean ± SD** Overall: 14.7 (6.4) IMI/REL: 14.6 ± 6.2 PIP/Taz: 14.8 >±6.7	IMI/REL is an appropriate treatment option for Gram-negative HABP/VABP, including critically ill, high-risk patients.
**Wunderink (2021) [31].**	**APEKS-NP Trial:** Phase 3 randomized, controlled, double-blind, parallel-group non-inferiority trial	IV 2 g CEF versus IV 2 g MERO q8 h for 7–14 days.	76 centers in 17 countries in Asia/Pacific, Europe, and North America. **Study date:** Oct 2017, and April 2019.	Hospitalized adult (≥18 y) patients with acute bacterial pneumonia in the form of HAP, VAP, or healthcare-associated Gram-negative pneumonia (HCAP). **Pathogen**: Gram-negative pathogens including *K. pneumoniae*, *P. aeruginosa*, *Acinetobacter baumannii*, *E. coli*, *E. cloacae*, and others.	n = 300 randomized, of which CEF, n = 148 (49%) MERO, n = 152 (51%)	**Modified ITT population (mITT) ^6^, n = 292:** CEF: n = 145 (49.7%) MERO: n = 147 (50.3%)	**Mean age ± SD:** CEF: 64.6 ± 14.6 MERO: 65.4 ± 15.1	**Admitted to ICU:** n = 199/292 (68%) **APACHE II; Mean ± SD** CEF: 16.0 ± 6.1 MERO: 16.4 ± 6.9	Cefiderocol was non-inferior to high-dose meropenem in terms of all-cause mortality on day 14 in patients with Gram-negative nosocomial pneumonia.
**Bassetti (2021) [32].**	**CREDIBLE-CR Trial**: Phase 3, randomized, open-label, multicenter, pathogen-focused, descriptive trial	IV 2 g CEF q8 h (+/−adjunctive treatment for pneumonia or sepsis) versus BAT ^ʊ^ for 7–14 days.	95 hospitals in 16 countries in North America, South America, Europe, and Asia. **Study date**: Sept 2016, and April 2019.	Hospitalized adult patients (≥18) with nosocomial pneumonia (HAP, VAP, HCAP), bloodstream infections or sepsis, or complicated UTI, and evidence of a carbapenem-resistant Gram-negative pathogen. **Pathogen**: Gram-negative pathogens including *Acinetobacter baumannii*, *K. pneumoniae*, *P. aeruginosa*, *S. Maltophilia*, *A. nosocomialis*, *E. cloacae*, and *E. coli.*	n= 152 randomized, of which CEF, n = 101 (66.4%) BAT, n = 51 (33.6%)	**ITT population and safety population ^7^, n = 150:** n = 101 cefiderocol (67.3%) n = 49 BAT (32.7%)	**Mean age ± SD:** CEF: 63.1 ± 19.0 BAT: 63.0 ± 16.7	**Admitted to ICU:** n = 78/150 (52%) **APACHE II; Mean ± SD** CEF: 15.3 ± 6.5 BAT: 15.4 ± 6.2	Cefiderocol had similar clinical and microbiological efficacy to best available therapy in infections caused by carbapenem-resistant Gram-negative bacteria.
**Kaye (2023) [33].**	**ATTACK Trial**: Phase 3, randomized trial This study had 2 arms—SUL-DUR A, and SUL-DUR B (observational arm)	IV 1 g/1 g SUL-DUR q6 h versus IV colistin (loading dose followed by 2.5 mg/kg q12 h) for 7–14 days All patients received IMI (1 g/1 g q6 h) as background therapy.	59 clinical sites in 16 countries in North America, Europe, Latin America, Asia/Pacific, and China. **Study date**: Sept 2019 to July 2021.	**Part A**: Hospitalized adult (≥18) patients with a confirmed diagnosis of HABP, VABP, ventilated pneumonia, or bloodstream infections known to be caused by ABC based on cultures. **Part B**: Same as above, but including complicated UTIs, acute pyelonephritis, and surgical or post-traumatic wound infections caused by ABC known to be resistant to colistin or polymyxin B.	n = 181 randomized (Part A): SUL-DUR (A), n = 92 COL (A), n = 89 n = 26 (Part B)	**Carbapenem-resistant ABC microbiologically modified ITT (CR-ABC-mMITT) ^8^, n = 156:** SUL-DUR (A): n = 64 Colistin: n = 64 Sul-DUR (B): n = 28	**Mean age (IQR):** SUL-DUR (A): 62 (54–75) Colistin: 66 (53–80) Sul-DUR (B): 59 (46–67)	**Admitted to ICU:** n = 45/156 (28.8%) **APACHE II; Mean ± SD** SUL-DUR (A): 16 ± 5 Colistin: 17 ± 5 SUL-DUR (B): 18 ± 5	SUL-DUR was non-inferior to colistin for the primary endpoint of 28-day all-cause mortality.

* Antibiotics: IV, intravenous; q6 h/q8 h, every 6/8 h; CEFT-TAZ, ceftolozane–tazobactam; CEF-AVI, ceftazidime–avibactam; MERO, meropenem; MERO-VAB, meropenem–vaborbactam; IMI/REL, imipenem–cilastatin–relebactam; PIP/TAZ, piperacillin–tazobactam; CEF, cefiderocol; SUL-DUR, sulbactam–durlobactam; BAT, best available therapy. Abbreviations: APACHE II, Acute Physiology and Chronic Health Evaluation II; VA(B)P, ventilator-associated (bacterial) pneumonia; HA(B)P, hospital-acquired (bacterial) pneumonia; ITT, Intention-To-Treat; cUTIs, complicated urinary tract infections; cIAIs, complicated intra-abdominal infections. Bacteria: ABC, Acinetobacter baumannii–calcoaceticus complex; *(K)lebsiella pneumoniae*; *(E)scherichia coli*; *(E)nterobacter cloacae* sp.; *(P)roteus mirabilis*; *(S)erratia marcescens; (S)tenotrophomonas Maltophilia*; *(A)cinetobacter nosocomialis*; CRE, carbapenem-resistant Enterobacteriaceae. ^Ƭ^ Paper provides numbers of patients ‘mechanically ventilated’ and ‘not ventilated’; however, it is not entirely clear what proportion of those are in ICU. ICU admission (% of specified population) and APACHE II scores at baseline (at randomization). ^Φ^ BAT included monotherapy/combination therapy with polymyxins, carbapenems, aminoglycosides, tigecycline, or ceftazidime–avibactam alone. ^ʊ^ BAT was pre-specified by the investigator before randomization and comprised of a maximum of three drugs.

### Specified Subgroups

^1^ Kollef 2019 [26], ITT group: all patients who received ≥1 dose of study drug.

^2^ Torres 2018 [27], clinically modified ITT population: all patients who met minimum disease criteria, with eligible Gram-negative pathogen, or those without any identifiable pathogen. 

^3^ Wunderink 2018 [28], microbiologically confirmed CRE infection (mCRE-MITT): all patients who received ≥1 dose of study drug and had lab-confirmed CRE. 

^4^ Motsch 2020, microbiologic modified ITT (mMITT) [29]: all patients with ≥1 dose of study drug and cultures confirming ≥1 qualifying pathogen (Gram-negative pathogen from the primary infection site). 

^5^ Titov 2021, modified ITT population (MITT) [30]: all patients with ≥1 dose of study treatment and whose baseline Gram stain did not show only Gram-positive cocci.

^6^ Wunderink 2020, modified ITT population (mITT) [31]: all patients who met criteria received at least one dose of study drug, excluding patients with Gram-positive monomicrobial infections.

^7^ Basetti 2020, ITT and safety population [32]: all patients that received treatment with study drug or best available therapy (BAT).

^8^ Kaye 2023, Carbapenem-resistant ABC microbiologically modified ITT (CR-ABC-mMITT) [33]: all patients with an isolated ABC organism that was confirmed to be carbapenem-resistant and receiving any amount of study drug. 

## 2. Ceftolozane–Tazobactam

In 2019, the FDA approved the use of ceftolozane–tazobactam for the treatment of hospital-acquired bacterial pneumonia (HABP), including ventilator-associated bacterial pneumonia (VABP) in adult patients (Table 1). Ceftolozane–tazobactam is a combination of an antipseudomonal fifth-generation cephalosporin with a β-lactamase inhibitor. Ceftolozane–tazobactam inhibits some Ambler class C β-lactamases, as well as a number of Ambler class A β-lactamases, more precisely most ESBLs, but not Klebsiella pneumoniae carbapenemases (KPCs). It is not active against class D (OXA) β-lactamases and metallo-β-lactamases (MBLs) [34]. 

Both ceftolozane and tazobactam are primarily renally excreted, and the dose must be adjusted for renal function (creatinine clearance [CrCl], ≤50 mL/min) (Appendix A, Appendix A). While the standard dose approved for intra-abdominal and urinary tract infections is 1.5 g every 8 h, a double dose of 3 g is recommended in nosocomial pneumonia, and is well tolerated. In a phase 1 clinical study in patients with VAP who received ceftolozane–tazobactam (3 g every 8 h), the lung penetration of the drug was adequate with a ratio of AUC in ELF-to-plasma of 50% for ceftolozane and 62% for tazobactam (Table 2). The PK/PD targets were achieved, as the mean concentration of ceftolozane and tazobactam remained above therapeutic levels in plasma for approximately 100% and 82% of the dosing interval, respectively, and in ELF for 100% of the dosing interval. These results were unchanged in patients with augmented renal clearance. [25]. Respectively, a high probability of target attainment was achieved in plasma and ELF with the high dose of ceftolozane–tazobactam, adjusted for renal function [35]. 

### 2.1. In Vitro Activity

Ceftolozane–tazobactam has potent antipseudomonal activity, and has been demonstrated in several studies to be the most active β-lactam against *P. aeruginosa* isolates [34,36]. Specifically, 96% of *P. aeruginosa* isolates (MIC ≤ 4 μg/mL) and 91.8% of *Enterobacterales* (MIC ≤ 2 μg/mL) from lower respiratory tract infections (LRTIs) were susceptible to ceftolozane–tazobactam, with 98% of the *Enterobacterales* species also being susceptible to meropenem. AmpC producers accounted for 70% of ceftolozane–tazobactam-resistant *Enterobacterales* [34].

In another study on patients hospitalized with pneumonia, ceftolozane–tazobactam inhibited 97.5% of *P. aeruginosa* isolates, 87.9% of MDR strains, 82.9% of XDR strains, and 90.8% of isolates that were resistant to meropenem. Amikacin and colistin were similarly effective. Of note, ceftolozane–tazobactam inhibited > 90% of all *Enterobacteriaceae* isolates, including 81.5% of ESBL, non-CRE isolates, compared to meropenem (96.3%) and piperacillin–tazobactam (74.1%) [37]. 

Comparative in vitro activity is depicted in Appendix A.

### 2.2. Clinical Trials 

In the ASPECT-NP trial, 726 patients with either VAP or ventilated HAP were randomized to treatment with ceftolozane–tazobactam or meropenem. A double dose of ceftolozane–tazobactam was used (2 g ceftolozane and 1 g tazobactam every 8 h with 1 h intravenous infusion). *Enterobacteriaceae* were isolated in 74% of patients, of whom 13% were resistant to ceftolozane–tazobactam, and *P aeruginosa* was isolated in 25%, of whom 3% were resistant to ceftolozane–tazobactam. One third of ESBL-producing *Enterobacteriaceae* were resistant to ceftolozane–tazobactam. Ceftolozane–tazobactam was shown to be non-inferior to meropenem in terms of 28-day all-cause mortality and clinical cure. In patients with ESBL-producing *Enterobacteriaceae*, 28-day mortality was lower in the ceftolozane–tazobactam treatment group (21%) compared with the meropenem group (29%). In addition, microbiological eradication for *Enterobacteriaceae*, ESBL-producing *Enterobacteriaceae*, and *P. aeruginosa* was similar among the two groups. The double dose of ceftolozane–tazobactam was demonstrated to be safe [26].

In a subgroup of patients with failure of the first line of antibacterial therapy, ceftolozane–tazobactam was associated with a significantly lower all-cause mortality compared to meropenem (22.6% vs. 45.0%) [38]. Additionally, in a subgroup analysis of patients with HAP/VAP caused by ESBL-positive and/or AmpC-producing *Enterobacterales*, 28-day all-cause mortality was also significantly lower in the ceftolozane–tazobactam treatment arm (6.7%) compared to the meropenem treatment arm (32.3%), while clinical cure and microbiological response rates were similar between the two groups. The *Enterobacterales* tested were carbapenemase-negative, while the most common ESBLs detected were of the CTX-M family, followed by SHV and OXA-1-like oxacillinases. Seven AmpC-overproducing isolates were observed (10,6%). All isolates were susceptible to ceftolozane–tazobactam and meropenem, and 30% were resistant to piperacillin–tazobactam. Further research in larger patient populations is warranted, to further delineate the reasons for the notable differences in mortality observed in patients taking ceftolozane–tazobactam, in the context of previous antibiotic treatment failure and in ESBL infections [39]. It should be noted that in the ASPECT-NP trial (Table 3), meropenem was administered in a relatively low dose (1 g every 8 h). However, the authors highlight that MIC values for meropenem were below 0.25 μg/mL for 77% of lower respiratory tract isolates, ensuring its efficacy [38].

A systematic review of 33 studies presenting ‘real world’ data for the treatment of nosocomial pneumonia with ceftolozane–tazobactam reported clinical success rates from 51.4 to 100.0% and microbiological success rates from 57.0% to 100.0%. *P. Aeruginosa* was the most common pathogen (92.8%) isolated [40].

Bassetti et al. studied HAP caused by ESBL-producing *Enterobacterales* and showed that clinical success was achieved in 78.3% of 46 patients treated with ceftolozane–tazobactam. The most frequent isolates were *Escherichia coli*, *Klebsiella pneumoniae*, and *Enterobacter* spp. The study supports the use of ceftolozane–tazobactam as empiric therapy when there is a high index of suspicion for ESBL infection, and supports its use as a carbapenem-sparing agent in areas where there is a high rate of resistance to carbapenem agents [41].

In patients with LRTIs caused by MDR/XDR *P. aeruginosa* strains, ceftolozane–tazobactam performs well when compared with conventional antibiotics. Clinical failure was significantly lower (by 73.3%) in the ceftolozane–tazobactam group, compared with the best alternative therapy group (including cefepime, meropenem, piperacillin–tazobactam, aminoglycosides, and polymyxins). According to microbiological data, the majority (87.4%) of MDR/XDR *P. aeruginosa* strains were susceptible to ceftolozane–tazobactam. In contrast, 79.6% were resistant to carbapenems, 11.4% to one or more aminoglycosides, and 20.6% to polymyxins. Adverse drug reactions, namely nephrotoxicity, encephalopathy, and neutropenia, were also less commonly reported in the ceftolozane–tazobactam group. [42]. Ceftolozane–tazobactam was also shown to be independently associated with the clinical cure in drug-resistant *P. aeruginosa* infections, 52% of which were cases of VAP [43]. Although in vitro susceptibility of *P. aeruginosa* to ceftolozane–tazobactam is reportedly similar to aminoglycosides and colistin in the literature [34,36,37], the outlined real-world data support possible clinical superiority of ceftolozane–tazobactam to aminoglycoside- and polymyxin-based regimens for the treatment of drug-resistant *P. aeruginosa*. 

The ASPECT-NP study reported the emergence of new resistance in 22.4% of patients in the meropenem treatment arm and in none of the patients in the ceftolozane–tazobactam arm [26].

On the other hand, it has been reported that exposure to ceftolozane–tazobactam resulted in 15–50% emergence of in vitro resistance in *P. aeruginosa*, and in addition, exposure to ceftolozane–tazobactam has been related to cross-resistance to ceftazidime–avibactam [44,45,46,47]. A 32-fold MIC increase was observed pre- and post-exposure to ceftolazone–tazobactam [44]. Development of resistance to ceftolazone–tazobactam is related to amino acid modifications in *Pseudomonas*-derived cephalosporinase enzymes (PDCs), following mutations and the over-expression of the bla AmpC genes. The administration of combination therapy did not restrict the emergence of resistance [48,49].

In conclusion, ceftolozane–tazobactam is valuable for treating nosocomial pneumonia caused by MDR *P. aeruginosa* strains. The experimental literature also supports a role for treating nosocomial pneumonia caused by ESBL-producing, carbapenemase-negative *Enterobacterales*, with the additional benefit of carbapenem-sparing and curbing the development of further resistance. However, this role is not currently supported in the guidelines, as the evidence base for clinical and microbiological benefit compared to carbapenem therapy is still not adequately established [47,48,49]. 

## 3. Ceftazidime–Avibactam

In 2018, ceftazidime–avibactam was approved by the FDA for the treatment of HABP and VABP (Table 1). 

Avibactam, a non-beta-lactam β-lactamase inhibitor, restores ceftazidime’s activity against certain β-lactamases. Specifically, ceftazidime–avibactam inhibits Ambler class A (e.g., ESBLs, KPCs), class C (e.g., AmpC), and some class D (e.g., OXA-48) β-lactamases [27], but is not active against class B metallo-β-lactamases (e.g., VIM, IMP, and NDM).

Satisfactory lung penetration was achieved in a phase 1 study with healthy volunteers, as AUC in ELF to plasma was ~31–32% for ceftazidime and ~32–35% for avibactam (Table 2). No serious adverse events were noted even with the high-dose group (3000 mg/1000 mg) [21]. The dose is adjusted for renal function (Appendix A, Appendix A). Although a post-antibiotic effect is not expected with ceftazidime–avibactam, it was observed in vivo in a neutropenic mouse lung infection model against *P. aeruginosa* strains [50].

### 3.1. In Vitro Activity

In an in vitro study, isolates from respiratory tract infections were tested for susceptibility to ceftazidime–avibactam. In total, 97% of *Enterobacter* spp., 98.5% of *Escherichia coli*, 94.7% of *Klebsiella pneumoniae*, and 91.2% of *Pseudomonas aeruginosa* were susceptible to ceftazidime–avibactam. Ceftazidime–avibactam exhibited potent antimicrobial activity even against >88% of MDR pathogens including *Enterobacter*, *Escherichia coli*, and *Klebsiella pneumoniae*—comparable to amikacin, tigecycline, colistin, and meropenem. However, ceftazidime–avibactam remained inferior to colistin for treatment of both MDR *Pseudomonas aeruginosa* (65.7% versus 97.1% of strains were susceptible, respectively) and XDR pathogens (<70% of XDR *E. coli* and XDR *P. aeruginosa* were susceptible to ceftazidime–avibactam). Only a minority (10%) of isolates were resistant to ceftazidime–avibactam, but of these, all were susceptible to colistin. *Pseudomonas aeruginosa* strains accounted for the majority of resistant isolates to ceftazidime–avibactam [51].

A study of *P. aeruginosa* isolates from respiratory tract infections in intensive care units in the United States showed that 91.7% of strains and 84.7% of carbapenem-resistant strains were susceptible to ceftazidime–avibactam [52]. A different study also found that 97% of *Pseudomonas aeruginosa* isolates were susceptible to ceftazidime–avibactam. Interestingly, this study found that 85.6% of piperacillin–tazobactam-resistant isolates and 84.8% of meropenem-resistant isolates remained susceptible to ceftazidime–avibactam, highlighting its potential role for infections with previous failure of antimicrobial treatment [53].

In the phase 3 randomized trial REPROVE, 98.6% of *Enterobacteriaceae* isolates were susceptible to ceftazidime–avibactam with MIC values below ≤8 μg/mL and 97.6% were susceptible to meropenem with MIC values below ≤1 μg/mL, while 88.4% of *P. aeruginosa* isolates were susceptible to ceftazidime–avibactam with MIC ≤ 8 μg/mL and 68.2% were susceptible to meropenem with MIC ≤2 μg/mL [54].

Comparative in vitro activity is depicted in Appendix A.

### 3.2. Clinical Trials

The phase 3 REPROVE study evaluated the efficacy of ceftazidime–avibactam (2 g of ceftazidime plus 0,5 g of avibactam) in nosocomial pneumonia compared with meropenem (Table 3). The most common isolates were *K. pneumoniae* and *P. Aeruginosa*. Ceftazidime–avibactam was non-inferior to meropenem in terms of the clinical cure at the test-of-cure visit (68,8% in the ceftazidime–avibactam group versus 73% in the meropenem group, *p* = 0.0066). All-cause mortality and microbiological eradication rates were similar between the two groups. Serious adverse events related to ceftazidime–avibactam were reported in 1% of patients. Concomitant aminoglycoside administration did not improve cure rates in either group of treatment [27].

Further trials are required to study the efficacy of ceftazidime–avibactam in pneumonia. Shields et al. reported 77 patients with CRE infections (of which 43% had pneumonia), treated with ceftazidime–avibactam. Ceftazidime–avibactam treatment resulted in clinical success in 55% of the combined patients, but only 36% of patients with pneumonia. These results were similar for monotherapy and combination therapy. Microbiological failure was noted in 32% of patients, while 10% of patients developed resistance that was observed mainly with KPC-3-producing *K. pneumoniae* infections. Of importance, clinical failure in this cohort was found to be associated with pneumonia and renal replacement therapy [55]. Furthermore, in a retrospective observational study, of 577 patients who were treated with ceftazidime–avibactam for carbapenemase-producing *K. pneumoniae* infections, a significant higher mortality was observed in patients with pneumonia, while a prolonged infusion of the drug, lasting ≥3 h, was associated with better survival. Similar to Shields’ study, combination therapy did not improve survival [56].

A number of studies have reported that ceftazidime–avibactam is superior to colistin for treatment of KPC-producing CREs [57]. One study in mechanically ventilated patients with CRE reported higher clinical cure and microbiological eradication rates, and highlighted ceftazidime–avibactam as an independent predictor for 28-day survival [58].

Onorato et al. report in their meta-analysis no statistically significant difference in mortality and microbiological cure in CRE infections between monotherapy and combination therapy (including with colistin, tigecycline, aminoglycosides, fosfomycin, and ciprofloxacin) [59].

Comparative clinical data among the new β-lactam–β-lactamase inhibitors are limited in the literature. Although ceftolazone is considered relatively more stable to hydrolysis by AmpC and exerts its potent antipseudomonal activity independently of tazobactam [48], ceftazidime–avibactam inhibits apart from AmpC β-lactamases, KPC, and OXA-48 serine β-lactamases, broadening its spectrum to include resistant *Enterobacterales*. In a retrospective cohort study comparing ceftolazone–tazobactam to ceftazidime–avibactam in MDR *P. aeruginosa* infections in Saudi Arabia, the clinical cure rate, in-hospital mortality, and 30-day mortality did not significantly differ among the two groups [60].

The emergence of resistance to ceftazidime–avibactam is a growing concern. Multiple mechanisms of resistance have been described. Mutations in the blaKPC genes prevent binding of ceftazidime–avibactam to its target site, due to amino acid changes in the omega loop of the KPC carbapenemase protein [48,61]. Furthermore, mutations in PDCs are responsible for the emergence of resistant strains of *P. aeruginosa* [48]. Exposure to ceftazidime–avibactam resulted in resistance emergence in 20% of *Enterobacterales* [62].

Moreover, a systematic review reported the emergence of resistance to ceftazidime–avibactam in KPC-producing *Enterobacterales* isolates. In the majority of cases, resistance was attributed to substitution mutation (D179Y) in KPC-3 and in KPC-2 enzymes [63]. Cross-resistance with ceftolazone–tazobactam is also reported in the literature, with strains that became resistant to ceftolazone–tazobactam after treatment also becoming resistant to ceftazidime–avibactam despite no prior exposure. In support of this, an 8-fold MIC increase to ceftazidime–avibactam was noted in patients treated with ceftolazone–tazobactam after the completion of treatment, while MICs for imipenem–relebactam were unchanged post-exposure [44]. Development of resistance to ceftazidime–avibactam does not seem to be affected by monotherapy or combination therapy use [48,64].

In conclusion, ceftazidime–avibactam may be used as a first-line therapy, as well as salvage therapy, in nosocomial pneumonia caused by CRE microorganisms (except for MBLs), given the high toxicity of alternative regimens and the emerging resistance to them. The optimization of treatment with continuous infusion is discussed in the literature [65]. The literature also highlights the importance of rapid CRE detection for the early initiation of appropriate antimicrobial treatment.

## 4. Meropenem–Vaborbactam

Meropenem–vaborbactam was approved in 2018 by the EMA for treatment of nosocomial pneumonia. Vaborbactam is a novel cyclic boronic acid β-lactamase inhibitor. Meropenem–vaborbactam is active against class A and class C β-lactamases, including KPCs, ESBLs, and AmpC. Vaborbactam does not inhibit class B MBLs and class D lactamases, namely OXA-48 [66]. As a result, the MICs of OXA-48 producers for meropenem remain unchanged with the addition of vaborbactam [67,68]. Its efficacy is not well demonstrated against strains of *P. aeruginosa* and *A.Baumanii*, where carbapenem resistance is not related to β-lactamase production [69].

Meropenem–vaborbactam is administered at a dose of 4 g (2 g meropenem/2 g vaborbactam) every 8 h as a 3 h prolonged infusion. The dosage is adjusted for renal function (Appendix A, Appendix A). Meropenem and vaborbactam exhibited satisfactory intrapulmonary penetration, with a mean AUC_0–8_ ELF in plasma ratio of 65% and 79%, respectively, in healthy volunteers [22] (Table 2).

### 4.1. In Vitro Activity

An in vitro study of 4790 *Enterobacterales* isolates from patients with HAP/VAP in the US showed that >99.9% were susceptible to meropenem–vaborbactam, while 98.7% were susceptible to amikacin, 97.2% to meropenem, and 96.6% to tigecycline. The majority (98.5%) of CRE isolates were susceptible to meropenem–vaborbactam and 96.9% to tigecycline, while susceptibility rates were lower for colistin, amikacin, and gentamicin (76.9%, 73.3%, and 52.7%, respectively). Resistant strains to meropenem–vaborbactam were NDM-1 and IMP producers. Concerning *P. aeruginosa*, meropenem–vaborbactam inhibited 89.5% of isolates, 59.0% of MDR, and 48.6% of XDR pathogens following colistin and amikacin, both of which showed more potent activity [70]. In a European study analyzing respiratory samples from patients hospitalized with pneumonia, 98.0% of *Enterobacterales* were susceptible to meropenem–vaborbactam, which inhibited 99.1% of KPC-producing isolates and 40.5% of OXA-48-like enzyme producers. It also inhibited 82.1% of *P. aeruginosa* strains, and 41.0% of MDR isolates [71].

A European study found that meropenem–vaborbactam was the least effective agent (compared to the new β-lactam–β-lactamase inhibitor combinations) for treating *P. aeruginosa* isolated from patients with pneumonia. Almost 88.7% of isolates were susceptible to meropenem–vaborbactam, compared to ceftolozane–tazobactam (93.3% were susceptible), imipenem–relebactam (94.3% were susceptible), and ceftazidime–avibactam (95.5% were susceptible). Colistin was the most active agent with 99.5% of strains being susceptible. Susceptibility rates were even lower for resistant strains [72].

Comparative in vitro activity is depicted in Appendix A.

### 4.2. Clinical Trials

In the TANGO II randomized clinical trial, meropenem–vaborbactam monotherapy was compared to the best available therapy in 75 patients with suspected CRE infections, of which 9.3% (7/75 patients) had HAP/VAP (Table 3). The best available therapy consisted of monotherapy or a combination of polymyxins, carbapenems, aminoglycosides, tigecycline, or monotherapy with ceftazidime–avibactam. Among 47 patients with confirmed CRE infections, higher clinical cure and microbiological eradication rates were achieved with meropenem–vaborbactam at the end of treatment (65.6% vs. 33.3%, *p*  =  0.03; and 65.6% vs. 40.0%, *p*  = 0.09, respectively). In patients with HAP/VAP or bacteremia, 28-day all-cause mortality was lower in the meropenem–vaborbactam group (*p*  =  0.25). Meropenem–vaborbactam was associated with fewer adverse events and lower nephrotoxicity rates compared to alternative therapies. Resistance to meropenem–vaborbactam was detected in five *K. pneumoniae* isolates with MIC  >  4 μg/mL and was related to metallo-beta-lactamases, class D carbapenemases (NDM or OXA-48), and one KPC-3, while one strain developed a ≥ 4-fold increase in MIC during treatment [28].

Shields et al. studied 20 patients with CRE infections, of which 6 patients had pneumonia (and of these, 5/6 had VAP). Clinical success was achieved in 67% of patients with pneumonia and 100% of patients with tracheobronchitis, while microbiological failure was noted in two patients with VAP. Renal replacement therapy did not affect these results [73]. Other studies have reported similar values, with clinical success achieved in 8/13 (61.5%) patients with serious Gram-negative pneumonia [74].

In a retrospective observational study, meropenem–vaborbactam was effective against KPC-producing *K. pneumoniae* infections, including those that were resistant to ceftazidime–avibactam. Especially, among patients with LRTIs, treated with meropenem–vaborbactam, in-hospital mortality was 30% for those attributed to KPC-producing *K. pneumoniae* and 20% for strains that were ceftazidime–avibactam-resistant [66].

Ackley et al. compared meropenem–vaborbactam to ceftazidime–avibactam in a retrospective study of 131 patients with CRE infections (of which 49 had pneumonia). Combination therapy was more often used in the ceftazidime–avibactam group (61.0% vs. 15.4%, *p*  <  0.01). Clinical success, day-30 and day-90 mortality, and adverse events were similar between the two groups. No emergence of resistance was observed in the meropenem–vaborbactam group, while three patients in the ceftazidime–avibactam monotherapy group with respiratory tract infections developed resistance, and in five patients, the MIC was increased. Two patients for whom initial therapy with ceftazidime–avibactam failed were subsequently successfully treated with meropenem–vaborbactam. The authors concluded that meropenem–vaborbactam might be associated with lower rates of emergent resistance for KPC-producing *Enterobacteriaceae*, compared to ceftazidime–avibactam, but the numbers were too small and statistical significance was not reached [64]. On the other hand, ceftazidime–avibactam exhibits a broader spectrum, also covering some OXA-48 producers and resistant *P. aeruginosa* strains [69,73].

Resistance of carbapenemase-producing *Enterobacterales* to meropenem–vaborbactam is usually associated with changes in permeability and efflux, due to mutations that result in porin loss [48,49].

In conclusion, meropenem–vaborbactam is a viable therapeutic option for pneumonia caused by KPC-producing CRE *Enterobacteriaceae*. Further studies are needed to identify the optimal agents to treat these serious infections; these need to consider both the clinical efficacy and emergence of resistance. Additionally, the rapid detection of KPC enzymes is essential to ensure the prompt initiation of the antibiotic, as delayed treatment beyond 48 h is associated with a worse clinical outcome [75].

## 5. Imipenem–Cilastatin–Relebactam (Imipenem–Relebactam)

In 2020, the FDA approved imipenem–cilastatin–relebactam (here-on referred to as imipenem–relebactam) for the treatment of HABP and VABP (Table 1). Relebactam is a non-β-lactam diazabicyclooctane (DBO) inhibitor. Imipenem–relebactam is active against Ambler class A (ESBLs, KPCs) and class C (AmpC) β-lactamases. It also shows activity against carbapenem-resistant *P. aeruginosa*, inhibiting *Pseudomonas*-derived cephalosporinases. Relebactam is inactive against Ambler class B carbapenemases [76], and like vaborbactam, does not improve imipenem’s MIC for OXA-48 carbapenemases [67,68,77], and consecutively does not restore susceptibility of OXA-48 producers to imipenem. *Morganella*, *Proteus*, and *Providencia* species have intrinsically decreased susceptibility to imipenem, and therefore also to imipenem–relebactam [78], while relebactam does not potentiate imipenem’s activity against *A.baumanii* [30].

The approved dose of imipenem–(cilastatin)–relebactam is 500/500/250 mg every 6 h as a 30 min intravenous infusion. Dose adjustment is required for patients with renal insufficiency [78] (Appendix A, Appendix A). Imipenem–relebactam exhibited satisfactory lung penetration in healthy subjects, with an AUC_0-∞_ in ELF to plasma of 54% for relebactam and 55% for imipenem after adjusting for protein binding (Table 2) [23].

### 5.1. In Vitro Activity

In a study examining lower respiratory tract samples from western Europe, 99.1% of *Enterobacterales*, 96.0% of piperacillin–tazobactam-resistant strains, and 81.1% of meropenem-resistant strains were susceptible to imipenem–relebactam. Imipenem–relebactam inhibited 91.4% of *P. aeruginosa*, 73.5% of piperacillin–tazobactam-resistant strains, and 40.5% of meropenem-resistant *P. aeruginosa* [76]. Studies in northern and Central Europe demonstrated that imipenem–relebactam was active against 99.6% of non-Morganellaceae *Enterobacterales* and 100% of ESBL-positive *E. coli*, *K. pneumoniae*, and *K. oxytoca*, while approximately 95% of all *P. aeruginosa* isolates and 43% of meropenem-resistant isolates were susceptible to imipenem–relebactam [79].

Imipenem–relebactam was the most active agent against 1445 isolates of *P. aeruginosa*, 32.8% of which were from respiratory tract infections (97.3% were susceptible), compared to colistin, ceftolozane–tazobactam, and ceftazidime–avibactam (approximately 94% were susceptible). The susceptibility rate of imipenem–relebactam for XDR strains was 86.4%. Thirty-seven isolates were resistant to imipenem–relebactam, with MIC > 8 μg/mL, due to produced carbapenemases’ VIM (n = 26), IMP (n = 4), and GES-5 (n = 7). Mainly, imipenem–relebactam retained its activity against 50% and 60.7% of strains that became resistant during therapy to ceftolozane–tazobactam and ceftazidime–avibactam, respectively [80].

A study reported that 58–59% of MDR *P. aeruginosa* that was resistant to ceftolozane–tazobactam remained susceptible to imipenem–relebactam. This is likely due to different mechanisms of developing resistance [81]. In support of this theory, imipenem–relebactam inhibited 37.1% of ceftolozane–tazobactam-resistant *P. aeruginosa* and ceftolozane–tazobactam inhibited 51.1% of imipenem–relebactam-resistant *P. aeruginosa* [79].

Zhang et al. collected 1886 *P. aeruginosa* isolates and 1889 *A. baumannii* isolates (65.6% and 72.4% from respiratory tract infections, respectively) in China. In total, 84.2% of *P. aeruginosa* and 65.8% of MDR strains were susceptible to imipenem–relebactam. Only 22.2% of *A.baumanii* and 5.3% of MDR *A. Baumanii* were susceptible to imipenem–relebactam [82].

Comparative in vitro activity is depicted in Appendix A.

### 5.2. Clinical Trials

RESTORE-IMI 1 was a phase 3, randomized trial comparing imipenem–relebactam with imipenem–colistin for the treatment of Gram-negative infections caused by imipenem-non-susceptible bacteria (Table 3). The microbiologic modified intent-to-treat population (mMITT) included 31 patients of whom 11 had HAP/VAP (8/21 in imipenem–relebactam group, and 3/10 in imipenem–colistin group). *P. Aeruginosa* was isolated in all patients with HAP/VAP. B-lactamases detected included AmpC, ESBLs, KPC, and OXA-48. A favorable overall outcome was noted in 87.5% of patients with HAP/VAP that received imipenem–relebactam and in 66.7% of those treated with imipenem–colistin. Overall, the results demonstrate that the imipenem–relebactam group had a superior 28-day favorable clinical response, and a lower rate of 28-day all-cause mortality and nephrotoxicity. Imipenem–relebactam was well tolerated [29].

The RESTORE-IMI 2 phase 3 randomized trial evaluated imipenem–relebactam compared to piperacillin–tazobactam in patients with HAP/VAP (Table 3). *K. pneumoniae* (25.6%), *P. aeruginosa* (18.9%), *Acinetobacter calcoaceticus–baumannii complex* (15.7%), and *Escherichia coli* (15.5%) were the most frequent pathogens isolated. Day-28 all-cause mortality was 15.9% in the imipenem–relebactam group and 21.3% in the piperacillin–tazobactam group, establishing non-inferiority. In fact, imipenem–relebactam was shown to have lower mortality rates in ventilated HAP/VAP and in critically ill patients with APACHE II scores ≥ 15 (Table 3). Favorable clinical response was achieved in 61.0% of patients treated with imipenem–relebactam and in 55.8% treated with piperacillin–tazobactam, while microbiological response was also similar between the two groups [30].

As with meropenem–vaborbactam, mechanisms of resistance to imipenem–relebactam include impermeability and porin loss, as well as increases in blaKPC copy numbers. *P. aeruginosa*’s resistance to imipenem–relebactam may arise as a result of an increased production of PDCs, loss of outer membrane porins (OprD), and overexpression of efflux pumps. Mutations in the MexAB-OprM and/or MexEF-OprN efflux operons have also been linked to the emergence of resistance of *P. aeruginosa* [48,49].

In conclusion, imipenem–relebactam is a viable treatment option for infections caused by difficult-to-treat, resistant *P. aeruginosa*, as well as CRE infections, excluding carbapenem-resistant *Enterobacterales* that produce MBLs (and imipenem-resistant OXA-48-like carbapenemase-producing strains). Imipenem–relebactam can also be employed as a rescue therapy in cases of resistance to ceftolozane–tazobactam and ceftazidime–avibactam.

## 6. Cefiderocol

Cefiderocol was approved by the FDA in 2020 for treatment of HABP/VABP (Table 1). Cefiderocol is a novel siderophore cephalosporin that inhibits all four Ambler classes of β-lactamases, including ESBLs, KPCs, MBLs (NDM, VIM, and IMP), and OXAs. It has activity against carbapenem-resistant *P aeruginosa*, carbapenem-resistant *A baumannii*, and *Stenotrophomonas maltophilia*. Cefiderocol has no activity against Gram-positive or anaerobic pathogens.

The dose of cefiderocol is 2 g, infused over 3 h, every 8 h, with dose adjustments for renal impairment or augmented renal clearance (Appendix A). In healthy subjects, ELF penetration was comparable to other β-lactams, with a penetration ratio of AUC_0–6_ in ELF to total plasma of 10.1%, and AUC_0–6_ in ELF to free (unbound) plasma of 23.9% (Table 2) [24]. In patients with pneumonia, the lung penetration ratio of cefiderocol was found to be 34% [83] (Table 2). In seven mechanically ventilated patients with pneumonia, cefiderocol penetration to the lung was confirmed, achieving a geometric mean ELF concentration of 7.63 mg/L at the end of the infusion and 10.4  mg/L at 2 h after the end of the infusion [20]. A prolonged infusion (3 h) is supported by the in vivo evaluation of cefiderocol in rat respiratory tract infection models against carbapenem-resistant *Pseudomonas aeruginosa*, *Acinetobacter baumannii*, and *Klebsiella pneumoniae* [84].

### 6.1. In Vitro Activity 

Cefiderocol has been shown to be the most active agent for inhibiting *CRE* (98.2%), compared to meropenem–vaborbactam, imipenem–relebactam, and ceftazidime–avibactam. It also inhibited 95.1%, 95.9%, and 89.2% of isolates that were resistant to meropenem–vaborbactam, imipenem–relebactam, and ceftazidime–avibactam, respectively. In total, 99% of *P. aeruginosa* and 97.3% of XDR isolates were susceptible to cefiderocol, while susceptibility rates for ceftazidime–avibactam, ceftolozane–tazobactam, and imipenem–relebactam ranged from 72% to 73%. Susceptibility of *A.baumanii* was 97.7% and that of meropenem-resistant isolates was 95.8%. Resistance to cefiderocol was detected in <1.5% of *Enterobacterales*, *P. aeruginosa*, *Acinetobacter species*, and *S. maltophilia* isolates [85].

Comparative in vitro activity is depicted in Appendix A.

### 6.2. Clinical Trials 

The phase 3 APEKS-NP study enrolled patients with Gram-negative nosocomial pneumonia. Cefiderocol was shown to be non-inferior to meropenem for the outcome of all-cause mortality at day 14 (Table 3). Both drugs were administered as a 3 h prolonged infusion and a high dose (2 g) of meropenem was prescribed. *K pneumoniae*, *P aeruginosa*, and *A baumannii* were the most common isolates. In total, 52% and 37% of *Enterobacterales* in the cefiderocol and meropenem group, respectively, were ESBL-positive, while 9% and 3% produced carbapenemases. In total, 70% and 63% of *A.baumanii* and 8% of *P. aeruginosa* in both groups produced carbapenemases. All-cause mortality on day 14 was 12.4% for the cefiderocol group and 11.6% for the meropenem group (*p* = 0.002). The clinical cure at test-of-cure and microbiological eradication were achieved in 65% and 48% in the cefiderocol group and 67% and 48% in the meropenem group. With regard to patients with *A.baumanii* pneumonia, mortality and clinical cure rates were similar between the two treatment groups. Both cefiderocol and meropenem in the APEKS-NP study were administered as monotherapy (Table 3) [31].

The phase 3 CREDIBLE-CR study evaluated cefiderocol in comparison with the best available therapy in carbapenem-resistant Gram-negative infections, 45% of which were nosocomial pneumonias (Table 3). *A.baumanii*, *Klebsiella pneumoniae*, and *P aeruginosa* were the most frequent carbapenem-resistant pathogens isolated. In patients with nosocomial pneumonia, the clinical cure rate at test-of-cure was 50% in the cefiderocol group and 53% in the best available therapy group, while microbiological eradication was achieved in 23% and 21% of patients, respectively. None of the patients in the cefiderocol group experienced recurrence of infection at follow-up, compared to 16% in the best available therapy group. All-cause mortality at day 14 was higher in the cefiderocol group (25% versus 11%) and this difference was mainly attributed to worse outcomes associated with *A.baumanii* isolation [32]. For this reason, there is a labeled warning in the product information highlighting the higher all-cause mortality that was observed in critically ill patients with carbapenem-resistant Gram-negative infections [86,87]. A similar proportion of patients in the two groups suffered treatment-emergent adverse events, but drug-related serious adverse events were noted in 1% in the cefiderocol group and 10% in the best available therapy group [32].

Cefiderocol was evaluated in comparison with colistin-based regimens, in 124 carbapenem-resistant *Acinetobacter baumannii* (CRAB) nosocomial infections, of which 28.5% represented cases of VAP. Although overall 30-day mortality was higher in patients treated with colistin (55.8% versus 34%, *p* = 0.018), for the subgroup of patients with VAP, 14-day and 30-day mortalities were similar between the two groups [88]. In another study evaluating cefiderocol in CRAB infections complicating severe COVID-19 pneumonia, 107 patients were analyzed, and 41% of them suffered from pneumonia. Cefiderocol was started a median of 2 days after the CRAB infection diagnosis, as monotherapy. Day 28 all-cause mortality was 55% in the cefiderocol group and 58% in the alternative treatment group [89]. Respectively, in an observational study by Russo et al., cefiderocol treatment was independently associated with 30-day survival in patients with COVID-19 complicated with VAP and bacteremia due to CRAB [90]. Finally, in a case series including 13 cases with CRAB pneumonia, cefiderocol treatment resulted in clinical cures in 53.8% of patients and in a 30-day mortality of 46%. It should be noted that in 38.5% of cases, cefiderocol was administered combined with other antibiotics, namely colistin [91].

Lower efficacy of cefiderocol in CRAB infections was documented in a PK/PD study, in which 11 patients with VAP caused by XDR *A. baumannii* were treated with cefiderocol at 2 g every 8 h in a 3 h infusion. All isolates were susceptible to cefiderocol and colistin. Microbiological failure was noted in 7/11 (63.6%) patients with VAP and was associated with suboptimal fCmin/MIC achieved in plasma. Therefore, the authors raise the question of increasing the dose of cefiderocol (e.g., 2 g every 6 h) administered to CRAB pneumonia to better attain the PK/PD target and maximize its efficacy [92].

In another prospective observational study, 23 patients with LRTIs received cefiderocol, with clinical failure rates of 31.3%, a 30-day microbiologic failure rate of 43.8%, and a 30-day mortality rate of 18.8%. Eighty-three percent (83.3%) of all pathogens were initially susceptible to cefiderocol and the most common pathogen was *P. aeruginosa*. The authors presume that these high failure rates were possibly related to underlying comorbidities [93]. A case series reported the use of cefiderocol (both as monotherapy and in combination with colistin, tigecycline, or aminoglycoside) as salvage therapy in carbapenem-resistant Gram-negative infections. Of 19 patients with pneumonia, the most common isolates included *Acinetobacter baumannii* and *P. aeruginosa*. Overall, 9/19 (47.4%) patients died, 3/19 (15.8%) patients experienced an infection recurrence, and clinical success was achieved in 7/19 (36.8%) patients [94].

Concerning *P. aeruginosa*, Meschiari et al. reported 17 patients with XDR and difficult-to-treat (DTR) resistant *P. aeruginosa* infections, who were treated with cefiderocol as rescue therapy after failure of previous antibiotic treatment and with no other existing alternative therapeutic option. Seven out of seventeen patients (41.2%) had VAP and one patient had nosocomial pneumonia. All the XDR/DTR *P. aeruginosa* strains were susceptible to cefiderocol, with MIC ≤ 2 mg/L. Clinical and microbiological cure rates were 62.5% (5/8) and 75% (6/8), respectively, while 3/8 (37.5%) died and 1/8 (12.5%) experienced a relapse. Notably, 7/8 patients received combination therapy with either colistin, ceftazidime–avibactam, or fosfomycin. [95].

Very recently, the results of the Perseus study (NCT05789199), a retrospective analysis (chart review) of real-world use of cefiderocol in the management of Gram-negative infections in Spain, were reported [96,97]. The analyzed cohort included 261 critically ill adult patients (63.2% in the ICU, 47.1% on mechanical ventilation, 28% with septic shock) that received cefiderocol consecutively for ≥72 h for a confirmed Gram-negative infection with limited treatment options: in 64.8%, the pathogens were resistant to all tested antibiotics and 44.4% had experienced treatment failure with prior treatment. Patients had serious comorbidities: immunosuppression (30.3%), solid or hematological tumors (23.8%), diabetes (22.2%), transplant (20.7%), chronic renal disease (13.0%), and chronic obstructive pulmonary disease (10.3%) [96,97]. The most frequent type of infection was respiratory tract infection (47.9%), and the most frequently isolated pathogens were *P. aeruginosa* (66.7%), *K. pneumoniae* (10.0%), and *S. maltophilia* (7.7%), while in 19.5% of cases, the infection was polymicrobial. The primary endpoint was defined as the composite of the clinical cure and/or survival at day 28, and was reported at 84.3%, while the day 28 all-cause mortality was reported as 21.5% [96,97].

Reports of the emergence of resistance to cefiderocol are uncommon. However, it is noteworthy and concerning that 15% of patients in the CREDIBLE-CR trial had a four-fold increase in the MICs of cefiderocol (despite the overall susceptibility to cefiderocol being maintained) [32]. Resistance mechanisms to cefiderocol include mutations in the TonB-dependent iron transport system affecting siderophore receptors and iron transport channels, mutations and amino acid alterations in AmpC β-lactamases, and increased expression of NDM or amino acid insertions in penicillin-binding protein-3 (PBP3) [15,48]. Mutations in the TonB-dependent iron transport system and amino acid changes in PDCs are also associated with resistance of *P. aeruginosa* strains [48]. The inactivation of the iron transporter gene cirA was found in NDM-producing *K. pneumoniae* that was highly resistant to cefiderocol and led to a nosocomial outbreak in a large tertiary care hospital in Italy [98]. Finally, cefiderocol heteroresistance was detected by Jacob E Choby et al., who analyzed carbapenem-resistant bacteria including *Acinetobacter baumannii*, *Klebsiella* spp., *Pseudomonas aeruginosa*, and *Stenotrophomonas maltophilia* [99]. This phenomenon of heteroresistance may, at least partially, explain the discrepancy between the results of APEKS-NP and CREDIBLE-CR trials [31,32,47].

In conclusion, cefiderocol is a new therapeutic option for patients with MDR Gram-negative nosocomial pneumonia caused by CRE and DTR *P. aeruginosa* and CRAB. Further studies are needed to investigate its role as a rescue/salvage therapy when other first-line antibiotic treatment has failed or resistance to other antibiotics has been found. As for *CRAB* infections, IDSA guidelines recommend cefiderocol only in combination with other active antibiotics [48]. The consensus statement of French experts suggests against the use of cefiderocol (‘probably not to be used’) for infections due to documented CRAB, unless there is unavailability of other treatment options [47].

## 7. Sulbactam–Durlobactam

In 2023, the FDA approved sulbactam–durlobactam for HABP and VABP caused by *Acinetobacter baumannii–calcoaceticus complex* (Table 1).

Sulbactam–durlobactam is the combination of the β-lactam sulbactam, which has intrinsic activity against *Acinetobacter* spp., with the next-generation diazabicyclooctane β-lactamase inhibitor durlobactam, which shields sulbactam from class A, C, and importantly class D β-lactamases. The dose is 1 g/1 g over a 3 h infusion every 6 h and the drug is excreted by the kidneys (Appendix A of Appendix A). For durlobactam, the mean ratio of AUC_0–6_ in ELF to total plasma was 37% and that for sulbactam was 50%, while for unbound plasma concentrations, these ratios were 41% and 81%, indicating a satisfactory intrapulmonary penetration [19] (Table 2).

Sulbactam–durlobactam shows promising in vitro efficacy against the *Acinetobacter baumannii–calcoaceticus* complex. In a study of 1722 isolates collected globally during 2016 and 2017, sulbactam–durlobactam inhibited 97.7% of *A. baumannii–calcoaceticus complex* isolates. Samples from respiratory tract infections represented 61.2% of the collection. Durlobactam lowered the MIC 90 of sulbactam by 32-fold. Only 46% were susceptible to meropenem, while 95.3% were susceptible to colistin, the only drug with susceptibilities comparable to sulbactam–durlobactam. The MIC 90 of sulbactam–durlobactam in respiratory tract infections was 2 μg/mL and that for MDR and XDR isolates was 4 μg/mL. Two percent (2.3%) of isolates were resistant to sulbactam–durlobactam with MIC above 4 μg/mL and resistance was attributed either to MBLs or amino acid changes in penicillin-binding protein-3 (PBP3), which is the target of sulbactam inhibition [100]. In another in vitro study performed recently, simulations for ELF exposure showed bactericidal activity of sulbactam–durlobactam against *A. baumannii–calcoaceticus* at an MIC 90 of ≤4 μg/mL [101]. A recent in vitro study performed in colistin-resistant and/or cefiderocol-resistant, carbapenem-resistant *A. baumannii–calcoaceticus* complex isolates from US hospitals showed that 89% of isolates were susceptible to sulbactam–durlobactam. The addition of imipenem in combination with sulbactam–durlobactam increased susceptibility to 97% with MIC 90 values’ cut-off being 4 and 8 μg/mL with and without imipenem co-administration [102].

### Clinical Trials

In the phase 3 ATTACK trial, sulbactam–durlobactam was non-inferior to colistin for the treatment of *Acinetobacter baumannii–calcoaceticus* complex (ABC) infections (Table 3). Both drugs were co-administered with imipenem, as background therapy, and the large majority of patients (176/181) suffered from HAP, VAP, or ventilated pneumonia. Sixty-nine percent (69%) of isolates were carbapenem-resistant. Day 28 all-cause mortality was 19% in the sulbactam–durlobactam group and 32% in the colistin group, while higher clinical cure rates were achieved with sulbactam–durlobactam. Nephrotoxicity was significantly more frequent in the colistin group [33].

In conclusion, sulbactam–durlobactam is a new weapon in our armamentarium against nosocomial pneumonia caused by resistant *Acinetobacter* spp. There are a few issues that need to be addressed by future studies. Firstly, and most importantly, its true efficacy needs to be replicated by real-life data rather than in the context of RCTs. Secondly, its administration as a monotherapy or as part of a combination regimen is controversial, given that in the ATTACK trial, imipenem was also given as a background therapy. Thirdly, the effect of combination therapy (e.g., with colistin or cefiderocol) on the outcome of infections remains to be determined. Finally, the efficacy of sulbactam–durlobactam or its combination with other antibiotics on other ABC infections apart from bacteremia or HAP/VAP remains to be answered.

## 8. Conclusions

Antimicrobial resistance represents an ever-increasing threat and an everyday challenge for clinicians, leading to delayed and/or inappropriate treatment and worse outcomes. During the past decade, several antibiotics with activity against MDR/XDR or DTR Gram-negative bacteria and HAP/VAP indication have been approved, with better safety profiles than the older ones (e.g., colistin) and have become the choice of treatment. Novel antibiotics mostly target specific mechanisms of resistance and show significant variability in terms of the spectrum and PK/PD. Ceftazidime–avibactam, meropenem–vaborbactam, and imipenem–relebactam are reserved for carbapenem-resistant *Enterobacterales* (CREs), while ceftazidime–avibactam, ceftolazone–tazobactam, and imipenem–relebactam are options for difficult-to-treat resistant (DTR) *P. aeruginosa*. Antibiotics covering MBL producers and carbapenem-resistant *A. baumanii* are still limited. Cefiderocol has the broadest in vitro spectrum of activity, including MBLs, *A. Baumanii*, and *Stenotrophomonas maltophilia* species; however, for *A. baumannii*, it is recommended to be administered in a combination, and also, more results from clinical trials are needed. Lastly, sulbactam–durlobactam is a promising option for *CRAB* pneumonia. Clinical data on comparative efficacy of the new β-lactam–β-lactamase inhibitors and cefiderocol are lacking in the literature. Future large, multicenter studies comparing clinical effectiveness and safety of the novel antibiotics are needed to define the preferred agents against MDR GNB pneumonia. Although the IDSA recommends three antibiotics (ceftazidime–avibactam, meropenem–vaborbactam, and imipenem–relebactam) equally for the treatment of CRE infections, the European guidelines prioritize ceftazidime–avibactam and meropenem–vaborbactam, as evidence for imipenem–relebactam is not adequate to recommend for or against its use [48,61].

Although the drug-related adverse events of novel antibiotics reported in randomized clinical trials are mild to moderate in severity, the long-term safety needs to be further confirmed as they are getting integrated in clinical practice. The emergence of resistance to ceftazidime–avibactam and ceftolazone–tazobactam is an increasing concern, as they are currently the most common agents studied in the literature and fewer reports refer to resistance arising with the other β-lactam–β-lactamase inhibitors and cefiderocol. Certainly, the documentation of developing resistance is an alarm to the unrestrictive use of novel antibiotics. Therefore, antimicrobial sensitivity testing (AST), and where possible, the identification of specific mechanisms of resistance, is of utmost importance for selecting the most appropriate antibiotic. The empirical administration of novel antibiotics may only be considered for patients with healthcare-associated, life-threatening pneumonia, who are at high risk for infection with carbapenem-resistant *Enterobacterales*, DTR *P. aeruginosa*, or *CRAB* (e.g., known colonization). In this context, local surveillance data are crucial to guide empirical treatment. Additionally, combination therapy of novel antibiotics with older agents, such as aminoglycosides, colistin, quinolones, and tetracyclines, has not been shown as more efficacious and is related to increased toxicity, when susceptibility to a β-lactam agent has been documented. On the contrary, at least two active agents against *CRAB* must be used, as efficacy of one single agent is questionable. In Table 4, a comparison between the IDSA and European guidance regarding the indications of novel antibiotics is depicted (excluding sulbactam–durlobactam that was approved in the publication of the guidance).

To conclude, novel antibiotics are valuable for the treatment of nosocomial pneumonia due to MDR GNB. However, it is crucial to utilize these agents appropriately, under specific criteria that are guided by local resistance patterns and surveillance, to improve nosocomial pneumonia clinical outcomes and, at the same time, to avoid the emergence of resistance and achieve a long shelf-life.

**Definitions**—CREs: IDSA defines them as members of the *Enterobacterales* order, resistant to >1 carbapenem antibiotic or producing a carbapenemase enzyme. Resistance to >1 carbapenem other than imipenem is required for bacteria generally not susceptible to imipenem, e.g., *Proteus* spp., *Morganella* spp., and *Providencia* spp. For the IDSA guidance, CREs refer to organisms displaying resistance to either meropenem or imipenem, or those *Enterobacterales* isolates producing carbapenemase enzymes. aDTR: IDSA defined it as *P. aeruginosa* exhibiting non-susceptibility to all of the following: piperacillin–tazobactam, ceftazidime, cefepime, aztreonam, meropenem, imipenem–cilastatin, ciprofloxacin, and levofloxacin. bDTR: ESCMID defined it as resistance to all b-lactams, including carbapenems, b-lactamase inhibitor combinations, and fluoroquinolones.

## Figures and Tables

**Figure 1 antibiotics-13-00629-f001:**
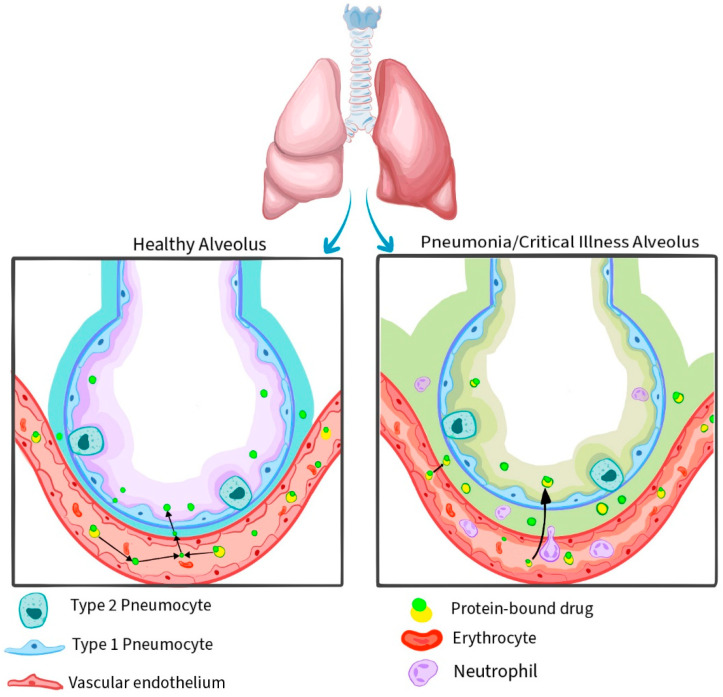
Physiological causes of differences in pharmacokinetic and pharmacodynamic parameters of antibiotics between normal alveoli and critical illness/pneumonia alveoli.

**Table 1 antibiotics-13-00629-t001:** Approved indications of the novel antibiotics included in this review.

Antibiotic	Approval	Indications
Ceftolazone–tazobactam	FDA initial approval 2014. FDA approval HAP/VAP 2019. EMA initial approval 2015. EMA approval HAP/VAP 2019.	FDA and EMA: HAP/VAP, cIAIs, cUTIs.
Ceftazidime–avibactam	FDA initial approval 2015. FDA approval HAP/VAP 2018. EMA initial approval 2016. EMA approval HAP/VAP 2016.	FDA and EMA: HAP/VAP, cIAIs, cUTIs. EMA: + associated bacteremia, aerobic GNB with limited treatment options.
Meropenem–vaborbactam	FDA approval 2017. EMA approval 2018.	FDA: cUTIs. EMA: HAP/VAP, cUTIs, cIAIs, associated bacteremia, aerobic GNB with limited treatment options.
Imipenem–relebactam	FDA initial approval 2019. FDA approval for HAP/VAP 2020. EMA approval 2020.	FDA and EMA: HAP/VAP. FDA: + cUTIs, cIAIs. EMA: + bacteremia associated with HAP/VAP, aerobic GNB with limited treatment options.
Cefiderocol	FDA initial approval 2019. FDA approval for HAP/VAP 2020. EMA approval 2020.	FDA: HAP/VAP, cUTIs. EMA: aerobic GNB with limited treatment options.
Sulbactam–durlobactam	FDA approval for HAP/VAP 2023.	FDA: HAP/VAP caused by susceptible strains of *Acinetobacter baumannii–calcoaceticus* complex.

cIAI: complicated intra-abdominal infection; cUTI: complicated urinary tract infection; GNB: Gram-negative bacteria; HAP: hospital-acquired pneumonia; VAP: ventilator-associated pneumonia; FDA: Food and Drug Administration (United States); EMA: European Medicines Agency.

**Table 4 antibiotics-13-00629-t004:** Comparison of the IDSA and European (ESCMID) guidelines [48,61].

IDSA 2023 Guidance on Gram-Negative Bacteria Infections ^	
Extended-spectrum β-lactamase-producing *Enterobacterales* (ESBL-Es)	The panel suggests preferentially preserving CEF-AVI, MERO-VAB, IMI-REL, and CEF for organisms exhibiting carbapenem resistance.
AmpC β-lactamase-producing *Enterobacterales* (AmpC-Es)	The panel suggests against the use of CEFT-TAZO for ESBL-E and AmpC-E infections, with the possible exception of polymicrobial infections.
Carbapenem-resistant *Enterobacterales* (CREs)	Where carbapenemase testing is not available or negative, CEF-AVI, MERO-VAB, and IMI-REL are preferred treatment options for CRE infections. In specific cases, CEF-AVI plus aztreonam, or CEF monotherapy, is recommended whilst awaiting antimicrobial sensitivity testing, and carbapenemase testing. These cases include (1) Patients with CRE infections who have received medical care in countries with a high prevalence of MBL-producing organisms in the past 12 months. (2) Patients with a culture positive for an MBL-producing isolate. Treatment of NDM/other MBL-producing infections: Panel recommends CEF-AVI in combination with aztreonam, or CEF as monotherapy. KPC-producing infections: Panel recommends MERO-VAB, CEF-AVI, and IMI-REL. CEF is an alternative option. OXA-48-like-producing infections: Panel recommends CEF-AVI. CEF is an alternative option. Note: Combination antibiotic therapy (i.e., a β-lactam agent with an aminoglycoside, fluoroquinolone, tetracycline, or polymyxin) is not suggested for the treatment of infections caused by CRE.
*Pseudomonas aeruginosa* with difficult-to-treat resistance (DTR *P. aeruginosa*)	CEFT-TAZO, CEF-AVI, and IMI-REL are preferred options for DTR *P. aeruginosa* infections. CEF is an alternative treatment option. For MBL-producing DTR *P. aeruginosa* isolates, the preferred treatment is CEF. Note: Combination antibiotic therapy is not suggested for infections caused by DTR *P. aeruginosa* if susceptibility to CEFT-TAZO, CEF-AVI, IMI-REL, or CEF has been confirmed.
Carbapenem-resistant *Acinetobacter baumannii* (CRAB)	Combination therapy with at least 2 active agents, wherever possible, is suggested for CRAB infections, at least until clinical improvement is observed. CEF should be limited to CRAB infections refractory to other antibiotics or in cases where intolerance or resistance to other agents precludes their use. When CEF is used, the panel suggests prescribing the agent as part of a combination regimen.
*Stenotrophomonas maltophilia*	Two main approaches are recommended for treatment of *S. maltophilia:* (1) Preferred therapy is with CEF as a component of combination therapy (with TMP-SMX, minocycline–tigecycline, or levofloxacin) at least until clinical improvement is observed. (2) Combination therapy with CEF-AVI and aztreonam, especially when critical illness is evident, or intolerance or inactivity of other agents is observed.
**ESCMID 2022 Guidelines for MDR GNB (endorsed by ESICM)**	
Third-generation cephalosporin-resistant *Enterobacterales* (3GCephREs)	Avoid use of new β-lactam–β-lactamase inhibitor (BLBLI) for 3GCephRE infections; reserve them for extensively resistant bacteria (good practice statement).
Carbapenem-resistant *Enterobacterales*	For severe infections due to CRE, the guidelines suggest MERO-VAB or CEF-AVI if active in vitro (conditional recommendation, moderate and low certainty of evidence). There is no evidence to recommend for or against the use of IMI-REL monotherapy for CRE. For severe infections due to MBL-producing CRE and/or organisms resistant to all other antibiotics (including CEF-AVI and MERO-VEB), the panel conditionally recommends treatment with CEF (conditional recommendation, low certainty of evidence) or aztreonam and CEF-AVI combination therapy (conditional recommendation, moderate certainty of evidence). For CRE infections susceptible to and treated with CEF-AVI, MERO-VAB, or CEF, the panel does not recommend combination therapy (strong recommendation against use, low certainty of evidence)
Carbapenem-resistant *Pseudomonas aeruginosa* (CRPA)	In patients with severe infections due to DTR-CRPA, the panel suggests therapy with CEFT-TAZO if active in vitro (conditional recommendation for use, very low certainty of evidence). Insufficient evidence is available for IMI-REL, CEF, and CEF-AVI. There is not enough evidence to recommend for or against the use of combination therapy with the new BLBLI (CEF-AVI and CEFT-TAZO) or CEF for CRPA infections.
Carbapenem-resistant *Acinetobacter baumannii* (CRAB)	For patients with CRAB susceptible to sulbactam and HAP/VAP, the panel suggests ampicillin–sulbactam (conditional recommendation, low certainty of evidence). Guidelines conditionally recommend against CEF for the treatment of infections caused by CRAB (conditional recommendations, low certainty of evidence) For patients with severe and high-risk CRAB infections, the guidelines suggest combination therapy including two in vitro active antibiotics among the available antibiotics (polymyxin, aminoglycoside, tigecycline, sulbactam combinations) (conditional recommendation, very low certainty of evidence).

NOTE: Sulbactam–durlobactam has not been approved by the FDA by the end date for which data were reviewed for the preparation of the guidance/guidelines. ^ Infections outside the urinary tract; the guidance document focuses on the treatment of infections in the United States. AmpC-Es: AmpC β-lactamase-producing Enterobacterales; AST: antibiotic susceptibility testing; BLBLI: β-lactam–β-lactamase inhibitor; CRAB: carbapenem-resistant Acinetobacter Baumannii; CREs: carbapenem-resistant Enterobacterales; CRPA: carbapenem-resistant Pseudomonas aeruginosa; ESBL-Es: extended-spectrum β-lactamase-producing Enterobacterales; ESCMID: European Society of Clinical Microbiology and Infectious Diseases; ESICM: European Society of Intensive Care Medicine; IDSA: Infectious Diseases Society of America; MBL: metallo-β-lactamase; KPCs: *K. pneumonia* carbapenemases; MDR: multi-drug resistant; DTR: difficult-to-treat resistance; NDMs: New Delhi metallo-β-lactamases; 3GCephREs: third-generation cephalosporin-resistant *Enterobacterales.*

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
