# Peer review of "Novel Antibiotics for Gram-Negative Nosocomial Pneumonia"

_antibiotics, 2024, doi:10.3390/antibiotics13070629_

Round 1

Reviewer 1 Report

Comments and Suggestions for Authors

The article addresses the rising concern of nosocomial pneumonia, especially those caused by multi-drug resistant Gram-negative bacteria (MDR-GNB), which are a major cause of death among critically ill patients. It examines new antibiotics developed to combat these resistant infections, assessing their effectiveness, pharmacokinetics, pharmacodynamics, and results from clinical trials. In general, it gives comprehensive well-written information, yet several things needs to be clarified.

1. Revise the citation format.

- Please change all the round brackets ( ) into square brackets [ ].

- The subsequent citation must be written in sequence. For example, [2], [3], [4] must be written [2-4].

2. Do the antibiotics mentioned in the article demonstrate comparable efficacy? The variability in clinical trial designs and outcome measures might complicate direct comparisons of different antibiotics' effectiveness.

3. Regarding the in vitro activity of the antibiotics againts the pathogenic microorganisms, please summarize the MIC data of the antibiotics in a table. Additionally, include detailed information on MIC for the antibiotics both as single agents and in combination with other antibiotics, such as β-lactamase inhibitors.

4. Based on the narrative reviews, it is suggested that authors add a section discussing the long-term impact of the antibiotics mentioned in this review. Is there any information available on the long-term safety or adverse effects of these new antibiotics?

5. Could the authors elaborate on the mechanisms through which these new antibiotics counteract resistance? 

Reviewer 2 Report

Comments and Suggestions for Authors

Comments for the authors:

This manuscript offers a good review of recent developments in antibiotics targeting nosocomial pneumonia, specifically focusing on multi-drug resistant (MDR) Gram-negative bacteria. Nosocomial pneumonia, including hospital-acquired pneumonia and ventilator-associated pneumonia, contunus to be a major cause of morbidity and mortality among critically ill patients. The authors highlight the growing prevalence of MDR Gram-negative bacteria, which complicates treatment and leads to extended hospital stays and high mortality rates.

The review covers the PK, PD, spectrum of antimicrobial activity, resistance patterns, and clinical efficacy of novel antibiotics approved since 2018. These antibiotics include ceftolozane-tazobactam, ceftazidime-avibactam, meropenem-vaborbactam, imipenem-cilastatin-relebactam, cefiderocol, and sulbactam-durlobactam. The authors highlight the importance of these antibiotics in combating MDR pathogens and note the emerging resistance to these new treatments. It is an information-rich manuscript, and I provided some questions and suggestions here for the authors’ consideration:

1. To improve the clarity and quality of English writing, consider having the manuscript reviewed by a native English speaker or a professional editing service to enhance readability.

2. This manuscript has a lot of typo and grammar errors. The authors should modify them very carefully. For example:

Line 20: Change "associated pneumonia is" to "associated pneumonia, is".

Line 23: Change "compounded by increased" to "compounded by the increased".

Line 24: Change "years, novel antibiotics" to "years, several novel antibiotics".

Line 35-37 in the Keywords: Remove the duplicate entry "nosocomial pneumonia".

Remove redundant acronyms for HAP and VAP as they are already defined below.

Standardize "multi-drug resistant" and "multidrug-resistance" to avoid repetition.

Line 57: Change "conventional aantibiotics" to "conventional antibiotics".

Comments on the Quality of English Language

The quality of English needs to be impoved to improve the clarity and enhance readability.
